# OpenReview forum: "AgentVocab: Structure-Aware Vocabulary Adaptation for Efficient LLM Agents"
_ICML.cc/2026/Conference — ICML 2026 regular_

### Official Review · Reviewer_RmBA · 2026-03-03

**Soundness:** 3
**Presentation:** 3
**Significance:** 3
**Originality:** 3
**Overall Recommendation:** 4
**Confidence:** 2

**Summary:**

The tokenizer for general LLMs struggles with structured tool-calling interactions, resulting in inefficient tokenization. Repetitive structural patterns and frequent semantic units in function calls are fragmented into lengthy sequences. To address this issue, this work introduces AgentVocab, a data-driven approach that enhances LLM efficiency by optimizing tokenization for tool calling. The experimental results show a 15-25% improvement on efficiency compared to the vanilla baseline, while maintaining tool-calling performance.

**Compliance With Llm Reviewing Policy:**

Affirmed.

**Key Questions For Authors:**

1. What's the relatively scales between structure tokens and content tokens, and what's the compression contributions/rations of these two types tokens?
2. In tool-calling datasets across different scenarios, what is the ratio of common tokens to specific tokens under AgentVocab? How efficient is it if only common tokens are added to the existing set?
3. How to handle the conflict of added tokens and existing tokens?
4. How should the method handle dynamically added APIs? For example, constructing token sets using only partial APIs from the training dataset or evaluating on benchmarks with APIs that differ from those in the training dataset.
5. What is the size of the added token set compare to the existing token set?

**Limitations:**

Limitation part is not explicitly described in paper.

**Strengths And Weaknesses:**

# Strengths:
1. AgentVocab enhances tokenization efficiency with a dual-branch strategy: structure-aware and content-aware token induction.
2. Experimental results demonstrate that AgentVocab significantly reduces decoding latency by 15-25% and decreases token length by 30%, which improves efficiency and stability in agent tool calling.
3. It offers practical value in agent training and can be easily applied on a case-by-case basis.

# Weaknesses:
1. Tool calling by agents can vary significantly across different scenarios. Each external token set is built from specific datasets, making it challenging to create a universal set.
2. To verify the robustness of this method, it is advisable to conduct additional experiments from various perspectives, such as different scenarios and base models with different tokenizers.
3. See questions.

---

> ### Author Rebuttal · Authors · 2026-03-31
>
> Thank you for your questions. We appreciate the opportunity to further clarify the points raised in our manuscript.
>
> > Weakness 1: Challenge of creating universal token sets across scenarios.
>
> We appreciate your insightful question regarding the challenge of building universal vocabularies. As this concern aligns closely with cross-task generalization and deployment costs, we kindly direct you to our detailed response to Reviewer 8BwP's Weakness 2. There, we elucidate how our domain-agnostic structural tokens naturally generalize to unseen tasks without requiring scenario-specific datasets.
>
> > Weakness 2: Robustness of this method.
>
> Thank you for asking about robustness of this method. For cross-tokenizer robustness via a Llama experiment, please refer to Reviewer 5XeX (Weakness 3 & Question 3).
>
> > Question 1: Relative scales and compression contributions.
>
> Thank you for asking about the scales and actual compression ratios. We explicitly designed the extraction ratio of structure-aware to content-aware tokens to be 1:1, mining 1,524 sequences for each branch. To quantify their respective contributions, we analyzed the trajectories generated by AgentVocab SFT. By calculating the product of each new token's frequency and its length savings, we observed that content tokens act as the primary engine for sequence compression, contributing approximately 80.1% of the total reduction. Structural tokens provide the remaining 19.9%. This confirms our design philosophy: content tokens maximize efficiency, whereas structural tokens are essential for ensuring syntactic accuracy in complex contexts.
>
> > Question 2: Distribution and independent efficacy of common versus specific tokens.
>
> Thank you for inquiring about cross-scenario common versus specific tokens. In our framework, common tokens correspond to structure-aware tokens, and specific tokens correspond to content-aware tokens. To rigorously investigate the efficiency of adding only common tokens, we compared the 2000-step Vanilla SFT against three 5000-step variants (each adding exactly 1524 tokens) using Qwen2.5-7B-Instruct fine-tuned on TouCan: a Generic BPE baseline, an Only-Content variant, and an Only-Structural variant.
>
> $\tau$-bench Results:
> | Model | Acc. | Input. | Output. | Lat. |
> | :--- | :--- | :--- | :--- | :--- |
> | Base | 13.94% | 7156.4 | 139.9 | 0.242 |
> | Vanilla SFT | 19.40% | 7553.6 | 137.0 | 0.235 |
> | BPE SFT | **18.79%** | 6992.3 | 127.8 | **0.198** |
> | AgentVocab SFT with Only Content Tokens | **16.36%** | 5186.9 | 118.6 | **0.179** |
> | AgentVocab SFT with Only Structural Tokens | **20.61%** | 5796.3 | 124.3 | **0.183** |
>
> $\tau^2$-bench Results:
> | Model | Acc. | Input. | Output. | Lat. |
> | :--- | :--- | :--- | :--- | :--- |
> | Base | 16.36% | 9252.4 | 154.9 | 0.404 |
> | Vanilla SFT | 21.56% | 9225.5 | 178.8 | 0.359 |
> | BPE SFT | **18.22%** | 8377.7 | 165.9 | **0.332** |
> | AgentVocab SFT with Only Content Tokens | **14.50%** | 6903.8 | 153.9 | **0.289** |
> | AgentVocab SFT with Only Structural Tokens | **18.96%** | 7836.7 | 180.7 | **0.341** |
>
> On complex tasks, Generic BPE and Content-only models suffer severe accuracy drops due to format hallucinations. Conversely, adding only common structural tokens demonstrates superior robustness, achieving the highest $\tau$-bench accuracy while explicitly improving latency over Vanilla SFT. Deploying only common structural tokens is highly efficient for maintaining syntactic validity across domains.
>
> > Question 3: Handling conflicts between added and existing tokens.
>
> We appreciate your question regarding potential token conflicts. Segmentation conflicts do not occur due to a definitive priority mechanism. Registered sequences gain absolute matching priority in the tokenizer. Before executing regular byte merges, the tokenizer performs a dictionary-based longest prefix match, seamlessly overriding the fine-grained segmentation path. It falls back to the base vocabulary only when matches fail.
>
> > Question 4: Handling dynamically added unseen APIs.
>
> Thank you for this critical deployment question. Our experimental setup inherently constitutes cross-domain validation. Our vocabulary is extracted exclusively from TouCan, while the evaluated benchmarks encompass entirely different domains with zero API overlap. AgentVocab stably achieves an approximate 25% compression rate because structural tokens are API-agnostic. Their invocation formats adhere strictly to standard JSON schemas, allowing peripheral syntax compression. For novel proper nouns, the tokenizer simply falls back to base subwords, guaranteeing zero parsing errors.
>
> > Question 5: Scale of vocabulary expansion and marginal parameter overhead.
>
> Thank you for inquiring about the expansion scale. The base model possesses 151,645 tokens. AgentVocab adds 3,006 tokens, yielding a marginal 1.98% expansion ratio. Increasing model parameters by less than 2% delivers nearly 30% context compression, demonstrating an extremely high return on investment.

---

> > ### Author Rebuttal · Reviewer_RmBA · 2026-04-01
> >
> > Thank you to the authors for your response. Most of my questions have been addressed, and I will keep the original score.

---

### Official Review · Reviewer_8BwP · 2026-03-10

**Soundness:** 4
**Presentation:** 4
**Significance:** 3
**Originality:** 4
**Overall Recommendation:** 5
**Confidence:** 4

**Summary:**

This paper addresses a key inefficiency in deploying large language models (LLMs) as agents: while LLMs are trained with general-purpose tokenizers designed for broad language coverage, their real-world agent usage is dominated by narrow, repetitive tool-calling interactions (e.g., function calls, JSON-formatted outputs). This mismatch causes common structural patterns to be fragmented into long sequences of low-level tokens, inflating decoding overhead. To solve this, the authors propose AgentVocab, a framework that mines real tool-calling traces to derive specialized vocabulary entries that better capture structural and semantic regularities in agent interactions — all without requiring task-specific schema engineering or architectural changes. Experiments on the τ-bench and τ²-bench benchmarks demonstrate that AgentVocab reduces inference latency by roughly 15–25% over a vanilla baseline while maintaining tool-calling accuracy, and the method is orthogonal to existing fine-tuning and agent-training approaches, making it easy to plug into standard agent pipelines.

**Compliance With Llm Reviewing Policy:**

Affirmed.

**Final Justification:**

My questions are fully resolved! Thank you very much!

**Key Questions For Authors:**

see weakness above

**Limitations:**

yes

**Strengths And Weaknesses:**

Strength:
1. The paper studies how the tokenization and vocabulary could affect efficiency and performance of language models. This sounds novel to me.
2. The paper is well-structured, with comprehensive analysis to understand the mechanisms behind

Weakness:
1. Is there results on other general benchmarks? Since the model is trained on a different vocabulary, I am wondering whether this could impact its capability on other tasks like coding, math, knowledge, etc. More importantly, it would be interesting to see whether the efficiency improvement in tool use is at the cost of performance degradation in other benchmarks.
2. Since there is a vocabulary change in order to make the proposed method work, does that mean every time we got a new task, we need to re-deign its vocabulary, collect in-domain data and then continue to train the model?

---

> ### Author Rebuttal · Authors · 2026-03-31
>
> Thank you for your questions. We appreciate the opportunity to further clarify the points raised in our manuscript.
>
> > Weakness 1: Impact of vocabulary expansion on general capabilities.
>
> We sincerely thank you for asking whether the efficiency improvements in tool use degrade the model's general reasoning capabilities. To objectively evaluate this, we conducted an additional assessment on the ARC scientific reasoning benchmark, a standard proxy for broad world knowledge and complex logic. Our evaluation rigidly follows the main experimental setup using Qwen2.5-7B-Instruct. We compared the un-finetuned Base model, the Vanilla SFT model (2000 steps), and our AgentVocab SFT model (5000 steps).
>
> | Model | ARC Overall | ARC-Easy | ARC-Challenge |
> |:---|:---|:---|:---|
> | Base | 90.61% | 91.96% | 87.88% |
> | Vanilla SFT | 89.97% | 91.33% | 87.20% |
> | AgentVocab SFT | 89.69% | 91.25% | 86.52% |
>
> The empirical results indicate that the Vanilla SFT model experiences a slight performance drop compared to the Base model, representing a common alignment tax associated with narrow-domain fine-tuning. Crucially, AgentVocab achieves scores virtually identical to the Vanilla SFT baseline. This explicitly demonstrates that modifying the tokenizer does not induce catastrophic forgetting or disrupt internal knowledge representations. By initializing new token embeddings via the average of their constituent subwords, the newly added tokens align seamlessly with the existing semantic space. The substantial efficiency gains in agentic tool use do not compromise fundamental reasoning skills.
>
> > Weakness 2: Necessity of redesigning vocabulary for new tasks.
>
> We appreciate your critical question concerning deployment costs, specifically whether new tasks require redesigning the vocabulary and collecting new in-domain data for retraining. In practice, such reconstruction is unnecessary.
>
> **The structural tokens we extract possess strong cross-domain universality.** Regardless of specific downstream tasks, mainstream interaction protocols share highly stable syntactic skeletons. Parameter passing based on JSON Schema follows the exact same structural patterns across entirely different business scenarios. Thus, structure-aware tokens are inherently domain-agnostic and generalize directly to unseen tasks without modification.
>
> **Relying solely on these universal structural tokens already yields significant benefits.** As demonstrated in our ablation studies, even without incorporating domain-specific content tokens, applying the universal structural vocabulary effectively prevents format hallucinations in long contexts and brings substantial improvements in inference efficiency and tool-calling accuracy.
>
> Developers can treat the universal structural vocabulary as a one-time foundational upgrade for language model agents to seamlessly handle most unseen downstream tasks. Extracting in-domain content tokens serves merely as an optional advanced optimization strategy reserved for highly vertical, high-traffic scenarios to maximize efficiency. Therefore, the core structural assets of AgentVocab are highly reusable and do not require frequent redesigns when shifting to new tasks.

---

> > ### Author Rebuttal · Reviewer_8BwP · 2026-04-01
> >
> > Thanks for the reply! My questions are fully resolved!

---

### Official Review · Reviewer_5XeX · 2026-03-11

**Soundness:** 3
**Presentation:** 3
**Significance:** 1
**Originality:** 2
**Overall Recommendation:** 4
**Confidence:** 4

**Summary:**

The paper studies LLM agents that repeatedly emit structured tool calls, where standard tokenizers break JSON/schema-heavy text into many small pieces. It proposes AgentVocab, which augments the base vocabulary with two types of new tokens: (i) structure-aware tokens, mined from spans built around JSON-schema/function-calling primitives, and (ii) content-aware tokens, mined from frequent token-level BPE merges ranked by expected token savings. The new vocabulary is added to the model, embeddings are initialized from constituent sub-tokens, and the whole model is then fine-tuned with standard SFT on tool-use trajectories.
The experiments use Qwen2.5-7B-Instruct fine-tuned on the Toucan SFT subset of about 119.3K tool interactions, and evaluate on τ-bench and τ²-bench. Relative to vanilla SFT, AgentVocab improves overall accuracy from 19.40% to 20.61% on τ-bench and from 21.56% to 21.93% on τ²-bench, while reducing average input tokens by about 26–30% and latency by about 25–29%. The paper also presents ablations suggesting structural tokens help more on correctness, while content tokens help more on efficiency.

**Compliance With Llm Reviewing Policy:**

Affirmed.

**Final Justification:**

the paper solved a practical challenge in agentic training. the proposed method is simple but yet effective.

**Key Questions For Authors:**

Can you compare against a generic vocabulary-adaptation baseline on the same Toucan corpus, with the same number of added tokens and matched fine-tuning compute? That is the most important missing experiment.

Can you run a compute-matched comparison where vanilla SFT gets the same effective training budget and checkpoint-selection procedure as AgentVocab? Right now 2k vs 5k muddies interpretation.

Can you show results on at least one additional backbone/tokenizer family? A Llama-family experiment would make the claim much stronger.

How much of the gain comes from the structure-aware branch specifically, as opposed to simply adding high-frequency new tokens from the tool corpus? A direct ablation against plain frequency/savings-based token addition would help.

Can you report tool-call validity / parseability / schema compliance explicitly? The paper talks about format failures and better stability, but a direct metric would be more convincing than qualitative discussion.

Please clarify the statement about 1,524 tokens vs 3,006 added tokens; the current wording is hard to parse.

**Limitations:**

yes

**Strengths And Weaknesses:**

Strengths
1. The problem is real. In deployed agents, structured tool schemas and repetitive argument patterns dominate usage, so tokenizer mismatch is a plausible and under-emphasized bottleneck. The paper frames this clearly and the method is easy to understand.
2. The efficiency gains are meaningful. A roughly 25–30% reduction in input tokens and around 25% latency reduction, with roughly preserved task success, is practically relevant for production agent systems. That is the paper’s strongest empirical point.
3. The paper also goes beyond a single aggregate table. It includes training-dynamics plots, structural-only vs content-only ablations, and a case study showing fewer redundant turns in a telecom scenario. Those analyses make the mechanism more interpretable than a bare before/after comparison.

Weakness

1. The main weakness is novelty relative to prior vocabulary-adaptation work. The paper’s closest neighbors are not general agent-training papers like Gorilla or ToolLLM, but prior work on extending vocabularies for efficiency in specialized distributions. AdaptiVocab already presents a full pipeline for domain-specific vocabulary modification, embedding initialization, tokenization patching, and adaptation training, and reports more than 25% token reduction without hurting quality. “Vocabulary Customization for Efficient Domain-Specific LLM Deployment” likewise extends a pretrained tokenizer with domain tokens, guarantees tokenization is never worse, and reports up to 20% shorter sequences and latency/throughput gains. VEGAD and HYPEROFA also contribute to the broader literature on post hoc vocabulary expansion and embedding initialization. Against that backdrop, AgentVocab’s most distinctive element is not “vocabulary adaptation for efficiency,” but the specific specialization to tool-calling traces plus the structure/content split. That is interesting, but it feels incremental rather than a clear conceptual jump.  Relatedly, the paper does not compare directly to the strongest relevant baselines. It compares against base Qwen and vanilla SFT, but not against a generic vocabulary-adaptation baseline on the same tool corpus, such as an AdaptiVocab-style savings-based tokenizer extension, or a tokenizer-extension method with no structure-aware branch. Without such baselines, it is hard to tell whether the gains come from the paper’s specific structure-aware design or simply from adding frequent new tokens from the tool corpus.

2. A second concern is experimental confounding. The vanilla model is trained for 2,000 steps, while AgentVocab is trained for 5,000 steps because the new embeddings need more learning. The paper gives a justification, but this still makes the comparison less clean than it should be. Since the main claim is that vocabulary adaptation improves both efficiency and behavior, I would want a tighter compute-matched comparison, or at least stronger evidence that the gains are not partly due to more training/selection effects.

3. A third concern is limited external validity. Everything is built on a single base model family, Qwen2.5-7B-Instruct, trained on one tool-use corpus and evaluated on two related benchmarks. There is no cross-backbone result showing the effect transfers to, say, Llama or another tokenizer family. That matters because tokenizer behavior and merge statistics are architecture/tokenizer dependent.

There is also a smaller but noticeable clarity issue in the vocabulary accounting. The paper says it “extract[s] 1,524 structural and content tokens” and that the union “results in the addition of 3,006 tokens,” which reads inconsistently. I suspect the intended meaning is 1,524 from each branch before union, but the current wording is confusing and should be fixed.

---

> ### Author Rebuttal · Authors · 2026-03-31
>
> Thank you for your questions. We appreciate the opportunity to further clarify the points raised in our manuscript.
>
> > Weakness 1 & Question 1: Comparison against a generic vocabulary-adaptation baseline.
>
> Thank you for highlighting the necessity of comparing against generic expansion methods. To validate our structure-aware design, we trained a strict generic baseline using Qwen2.5-7B-Instruct on TouCan. We applied the standard savings-based BPE algorithm globally, extracting exactly 3,006 high-frequency tokens. This BPE SFT model was trained for 5,000 steps to match AgentVocab's compute budget. Results on $\tau^2$-bench are as follows:
>
> | Model | Steps | Acc. | Input. | Output. | Lat. |
> | :--- | :--- | :--- | :--- | :--- | :--- |
> | Base | 0 | 16.36% | 9252.4 | 154.9 | 0.404 |
> | Vanilla SFT | 2000 | 21.56% | 9225.5 | 178.8 | 0.359 |
> | BPE SFT | 5000 | **15.99%** | 8415.3 | 172.3 | **0.340** |
> | AgentVocab SFT | 5000 | **21.93%** | 6795.2 | 149.7 | **0.302** |
>
> While BPE SFT reduces sequence lengths, its accuracy plummets to 15.99%. This confirms that syntax-blind BPE indiscriminately merges tokens across JSON boundaries, triggering severe format hallucinations during long-context generation. Conversely, AgentVocab's paradigm protects syntactic integrity, achieving the highest accuracy and maximizing efficiency.
>
> > Weakness 2 & Question 2 & Question 5: Compute-matched comparison and fine-grained compliance metrics.
>
> Thank you for raising concerns regarding experimental fairness and format stability.  Please refer to our detailed response to Reviewer TVXa (Weakness 1 & Question 1) for the full analysis.
>
> > Weakness 3 & Question 3: Cross-backbone generalization on Llama family.
>
> We appreciate your suggestion to evaluate across tokenizer families. To verify cross-architecture robustness, we fine-tuned Llama-3.2-1B-Instruct on TouCan. Combining our tokenizer-agnostic 1,524 structural tokens with 1,524 Llama-specific content tokens yielded 3,002 unique additions after deduplicating 46 overlapping items. Evaluated on $\tau^2$-bench:
>
> | Llama-3.2-1B Config | Steps | Acc. | Input. | Output. | Lat. |
> | :--- | :--- | :--- | :--- | :--- | :--- |
> | Base | 0 | 8.18% | 19030.2| 1264.8 | 1.138 |
> | Vanilla SFT | 1500 | 8.55% | 9963.5 | 340.1 | 0.351 |
> | Vanilla SFT | 2000 | 6.32% | 10417.3| 329.3 | 0.404 |
> | AgentVocab SFT | 2000 | **8.92%** | 10258.9| 507.7 | **0.581** |
>
> AgentVocab's latency trails the 1500-step Vanilla baseline, attributable to a 1B-parameter model's capacity constraints in adequately optimizing 3,000 new embeddings. However, a critical stability threshold emerges: small models are highly vulnerable to context fragmentation. Vanilla SFT suffers severe semantic degradation at 2,000 steps. Conversely, AgentVocab successfully averts format collapse, securing the highest accuracy.
>
> > Weakness 4 & Question 6: Clarity of vocabulary accounting.
>
> We are grateful for your careful reading. We independently extracted 1,524 tokens for each branch. Due to an inherent overlap of 42 tokens, their union results in exactly 3,006 added tokens. We will correct this phrasing in Section 4.1 to eliminate ambiguity.
>
> > Question 4: Ablation isolating the structure-aware branch versus generic frequency addition.
>
> Thank you for requesting this direct ablation. Using Qwen2.5-7B-Instruct fine-tuned on TouCan, we compared the 2000-step Vanilla baseline against three variants trained for 5000 steps, each adding exactly 1,524 tokens: a Generic BPE baseline , an Only-Content variant , and an Only-Structural variant.
>
> $\tau$-bench Results:
> | Model | Acc. | Input. | Output. | Lat. |
> | :--- | :--- | :--- | :--- | :--- |
> | Base | 13.94% | 7156.4 | 139.9 | 0.242 |
> | Vanilla SFT | 19.40% | 7553.6 | 137.0 | 0.235 |
> | BPE SFT | **18.79%** | 6992.3 | 127.8 | **0.198** |
> | AgentVocab SFT with Only Content Tokens | **16.36%** | 5186.9 | 118.6 | **0.179** |
> | AgentVocab SFT with Only Structural Tokens | **20.61%** | 5796.3 | 124.3 | **0.183** |
>
> $\tau^2$-bench Results:
> | Model | Acc. | Input. | Output. | Lat. |
> | :--- | :--- | :--- | :--- | :--- |
> | Base | 16.36% | 9252.4 | 154.9 | 0.404 |
> | Vanilla SFT | 21.56% | 9225.5 | 178.8 | 0.359 |
> | BPE SFT | **18.22%** | 8377.7 | 165.9 | **0.332** |
> | AgentVocab SFT with Only Content Tokens | **14.50%** | 6903.8 | 153.9 | **0.289** |
> | AgentVocab SFT with Only Structural Tokens | **18.96%** | 7836.7 | 180.7 | **0.341** |
>
> The empirical data isolates performance gains. **Generic BPE and Content-only models achieve lower latency but suffer severe accuracy degradation.** Pure frequency-driven algorithms are syntax-blind, indiscriminately truncating JSON boundaries and triggering format hallucinations. **Conversely, adding only structural tokens maintains robust accuracy while explicitly improving latency over Vanilla SFT.** This proves encoding protocol skeletons into atomic units safeguards syntactic validity, whereas frequency tokens primarily drive sequence compression.

---

> > ### Author Rebuttal · Reviewer_5XeX · 2026-04-03
> >
> > my questions are resolved.

---

### Official Review · Reviewer_TVXa · 2026-03-19

**Soundness:** 2
**Presentation:** 3
**Significance:** 2
**Originality:** 3
**Overall Recommendation:** 4
**Confidence:** 3

**Summary:**

This paper studies a training-deployment mismatch for LLM agents: general-purpose tokenizers are used in deployments dominated by repetitive, structured tool-calling traces. The authors propose AgentVocab, which augments a base vocabulary with structure-aware spans mined from schema-like patterns and content-aware tokens mined via token-level BPE, then fine-tunes the model with the adapted tokenizer. Experiments on $\tau$-bench and $\tau^2$-bench report sizable reductions in token counts and latency, with roughly preserved or slightly improved pass rates relative to a vanilla SFT baseline.

**Compliance With Llm Reviewing Policy:**

Affirmed.

**Final Justification:**

my questions are resolved.

**Key Questions For Authors:**

1. Can you provide a strictly controlled comparison with matched training budget and checkpoint selection?
This is the most important issue for me. Please compare Vanilla SFT and AgentVocab SFT under equal numbers of update steps, equal seen examples, or ideally equal training FLOPs. Also specify a checkpoint-selection rule that does not advantage one method post hoc. If the efficiency and accuracy trends still hold under a fair control, my confidence in the paper would increase substantially.
2. Can you decompose the efficiency gains into tokenization compression versus behavioral trajectory shortening?
Figure 4 suggests a large part of the input-token reduction comes from fewer interaction turns, not only shorter tokenization. Please quantify:
token-count reduction when retokenizing the exact same trajectories with the new tokenizer, and additional reduction due to changed agent behavior.

**Limitations:**

yes

**Strengths And Weaknesses:**

Strengths:

1. The paper tackles a real systems problem that is easy to overlook.
The central motivation, laid out in Sections 1 and 3.1, is sensible: agent deployments often generate lots of JSON-like tool calls and schema-heavy context, yet they inherit tokenizers built for general web text.
2. The method is conceptually simple and reasonably well motivated. The decomposition into structural tokens and content tokens is intuitive.
3. The empirical efficiency gains are substantial in the reported setup. Table 1 and Table 2 both show large reductions in input tokens, output tokens, and latency. On the face of the reported numbers, the efficiency story is strong.
4. The main paper is organized in a standard way, and the narrative flow from motivation to method to experiments is easy to follow. For a systems-oriented paper, this matters.

Weaknesses：

1. The main comparison is not controlled fairly, and this undermines the core empirical claim.
This is my biggest concern. In Section 4.1, the vanilla model is trained for 2,000 steps, while AgentVocab is trained for 5,000 steps. The authors justify this by saying the adapted-vocabulary model needs extra time to learn new embeddings, and Figure 3 is then used to claim that the vanilla model is best around earlier checkpoints. But this is still not a clean apples-to-apples comparison. If two systems differ in tokenizer, training budget, and checkpoint selection policy, then it becomes difficult to attribute the accuracy or stability differences to vocabulary adaptation itself.
This matters because a central claim of the paper is not just improved efficiency from shorter tokenization, but also preserved or slightly improved tool-calling performance and stability. With unequal optimization budgets, one cannot cleanly separate “better tokenizer” from “more training” or “more favorable checkpoint choice.” A fairer protocol would compare equal compute, equal number of seen examples, or at least matched checkpoint sweeps for both conditions with a principled selection rule fixed in advance.

2. The originality is modest relative to the breadth of the claims.
The core recipe is: mine frequent structured spans, mine frequent content merges by a savings metric, add them to the vocabulary, initialize via average of constituent embeddings, and fine-tune. That is a reasonable engineering combination, but the paper’s more ambitious language about “a critical optimization” for agentic systems feels ahead of the actual methodological novelty.
I do think the specific application to tool-calling agents is useful. But as a method contribution, this reads more like a targeted adaptation of existing vocabulary-customization ideas to a new use case than a fundamentally new technique.

---

> ### Author Rebuttal · Authors · 2026-03-31
>
> Thank you for your questions. We appreciate the opportunity to further clarify the points raised in our manuscript.
>
> > Weakness 1 & Question 1: Matched compute comparison and checkpoint selection.
>
> We sincerely thank you for evaluating our experimental fairness regarding matched training budgets. To eliminate confounding factors, we trained Vanilla SFT and AgentVocab SFT models (based on Qwen2.5-7B-Instruct) on the TouCan-SFT dataset with identical hyperparameters from Step 0 to 5000 continuously. Evaluated on $\tau^2$-bench, we introduced three metrics: Parseability for valid JSON decoding, Tool Validity for zero hallucinated tools, and Schema Compliance for adherence to constraints.
>
> | Model | Steps | Accuracy | Latency | Parseability | Validity | Schema Compliance |
> |:---|:---|:---|:---|:---|:---|:---|
> | Base | 0 | 16.36% | 0.404 | 99.68% | 90.42% | 90.01% |
> | Vanilla SFT | 2000 | 21.56% | 0.359 | 99.95% | 97.79% | 97.79% |
> | Vanilla SFT | 5000 | N/A* | 0.644 | 100.0% | 83.81% | **83.16%** |
> | AgentVocab SFT | 5000 | 21.93% | 0.302 | 99.71% | 98.71% | **97.55%** |
>
> *Vanilla SFT at 5000 steps produced severe schema violations interrupting automated evaluation.
>
> Forcibly trained to 5000 steps, Vanilla SFT suffers severe **overfitting** to surface-level syntactic patterns. Despite 100% parseability, Validity and Schema Compliance drop to approximately 83%, underperforming the Base model. It depletes capacity fitting tedious JSON punctuation, degrading high-level reasoning. Conversely, AgentVocab exhibits exceptional stability at 5000 steps with 98.71% validity and 0.302s latency. This validates our **dynamic early-stopping rule**: selecting the lowest latency checkpoint once accuracy plateaus. Vanilla peaks at 2000 steps, while AgentVocab requires 5000 steps to semantically align new embeddings. Arbitrary truncation fails to reflect true upper bounds.
>
> > Weakness 2: Methodological originality and system-level contributions.
>
> We appreciate your objective assessment of the engineering utility. Our primary motivation is resolving the fundamental distributional mismatch between general-purpose tokenizers and structured agentic interactions. As tool-calling forms the foundation of modern agent systems, this mismatch severely degrades context efficiency and logical coherence. AgentVocab achieves architecture-agnostic system-level optimization via vocabulary adaptation.
>
> Algorithmically, AgentVocab is not a naive application of frequency-driven BPE. Pure data-driven methods indiscriminately truncate JSON boundaries, causing format collapse. We address this via a dual-branch paradigm decoupling structure from content. In the structural branch, we abandon traditional frequency merging, designing a contiguous span extraction algorithm based on structural primitives. This ensures complex protocol skeletons are extracted intact as indivisible units, preserving format integrity.
>
> > Question 2: Quantitative decomposition of efficiency gains.
>
> Thank you for the constructive question regarding efficiency decomposition. We cross-evaluated complete trajectories generated by Vanilla SFT at 2000 steps and AgentVocab at 5000 steps on $\tau^2$-bench using both tokenizers. Average total tokens per interaction:
>
> | Tokenizer Used | Vanilla SFT Trajectory | AgentVocab SFT Trajectory |
> |:---|:---|:---|
> | Base Tokenizer | **7558.5** | 6994.1 |
> | AgentVocab Tokenizer| 5390.8 | **5052.2** |
>
> AgentVocab's total reduction of 2506.3 tokens (7558.5 to 5052.2) is precisely decomposed. First, controlling the text to remain unchanged using the Vanilla trajectory, replacing the Base tokenizer with AgentVocab drops token consumption to 5390.8. The innate compression of the specialized vocabulary directly eliminates 2167.7 redundant tokens (**86.49%** of total gain), independent of inference behavior.
>
> Second, uniformly applying the AgentVocab tokenizer, the AgentVocab trajectory measures 5052.2 tokens, shorter than the Vanilla trajectory at 5390.8. The model effectively bypasses redundant retry loops caused by format hallucinations. This compact behavioral pattern contributes an additional reduction of 338.6 tokens (**13.51%** of total gain). Efficiency improvements are thus jointly driven by high-ratio vocabulary compression and optimized interactive logic.

---

### Decision · Program_Chairs · 2026-04-30

**Decision:**

Accept (regular)

**Comment:**

Summary:
This paper introduces AgentVocab, a structure-aware vocabulary adaptation framework for efficient LLM agents. The authors fine-tuned a specialized vocabulary with structure-aware spans mined from schema patterns and show that it works effectively on the tau-bench and tau^2-bench reports.
Justifications:
All reviewers agree that the paper addresses a real problem, demonstrating novelty and practical value. The proposed method is simple yet effective. The authors provided detailed rebuttals that address most of the reviewers' concerns and questions. I recommend incorporating this feedback and additional experiments from the rebuttal into the final version of the paper.